# RETRACTED: Effect of *Bifidobacterium bifidum* Supplementation in Newborns Born from Cesarean Section on Atopy, Respiratory Tract Infections, and Dyspeptic Syndromes: A Multicenter, Randomized, and Controlled Clinical Trial

**DOI:** 10.3390/microorganisms12061093

**Published:** 2024-05-28

**Authors:** Anna Rita Bellomo, Giulia Rotondi, Prudenza Rago, Silvia Bloise, Luigi Di Ruzza, Annamaria Zingoni, Susanna Di Valerio, Eliana Valzano, Francesco Di Pierro, Massimiliano Cazzaniga, Alexander Bertuccioli, Luigina Guasti, Nicola Zerbinati, Riccardo Lubrano

**Affiliations:** 1Dipartimento Materno Infantile e di Scienze Urologiche, Sapienza Università di Roma, UOC di Pediatria e Neonatologia-Polo Pontino, 04100 Latina, Italy; annaritabellomo62@gmail.com (A.R.B.); prudenzarago@gmail.com (P.R.); 2Pediatric Surgery Unit, Gaslini Children Hospital and Research Institute, 16147 Genoa, Italy; 3UOC Pediatria e Nido, Ospedale S.S. Trinità, 03039 Sora, Italy; 4UOC Pediatria e Neonatologia, Ospedale G.B. Grassi, 00122 Ostia, Italy; 5UOC Neonatologia e Terapia Intensiva Neonatale, Ospedale S. Spirito, 65124 Pescara, Italy; 6Scientific & Research Department, Velleja Research, 20125 Milan, Italy; 7Department of Medicine and Surgery, University of Insubria, 21100 Varese, Italy; 8Department of Biomolecular Sciences, University of Urbino Carlo Bo, 61029 Urbino, Italy; alexander.bertuccioli@uniurb.it

**Keywords:** children, dyspepsia, microbiota analysis, probiotics, 16S rRNA, ITS

## Abstract

Cesarean section is considered a possible trigger of atopy and gut dysbiosis in newborns. Bifidobacteria, and specifically *B. bifidum*, are thought to play a central role in reducing the risk of atopy and in favoring gut eubiosis in children. Nonetheless, no trial has ever prospectively investigated the role played by this single bacterial species in preventing atopic manifestations in children born by cesarean section, and all the results published so far refer to mixtures of probiotics. We have therefore evaluated the impact of 6 months of supplementation with *B. bifidum* PRL2010 on the incidence, in the first year of life, of atopy, respiratory tract infections, and dyspeptic syndromes in 164 children born by cesarean (versus 249 untreated controls). The results of our multicenter, randomized, and controlled trial have shown that the probiotic supplementation significantly reduced the incidence of atopic dermatitis, upper and lower respiratory tract infections, and signs and symptoms of dyspeptic syndromes. Concerning the gut microbiota, *B. bifidum* supplementation significantly increased α-biodiversity and the relative values of the phyla Bacteroidota and Actinomycetota, of the genus *Bacteroides*, *Bifidobacterium* and of the species *B. bifidum* and reduced the relative content of *Escherichia*/*Shigella* and *Haemophilus*. A 6-month supplementation with *B. bifidum* in children born by cesarean section reduces the risk of gut dysbiosis and has a positive clinical impact that remains observable in the following 6 months of follow-up.

## 1. Introduction

It is a widely shared opinion that the increase in allergy rates observed in highly developed countries, referable to the last few decades, relates to alterations in the structure of the intestinal microbiota [1]. These early alterations in the gut bacterial consortium are essentially attributable to a lower α-biodiversity, accompanied by a reduced relative presence of genera considered “useful” at the beginning of life, such as *Bifidobacterium* and *Bacteroides*, and by an increased presence of Gram-negative species, such as those belonging to the phylum Pseudomonadota (former Proteobacteria), or considered not eubiotic like staphylococci and clostridia [2]. Particularly beneficial properties are attributable to the genus *Bifidobacterium*, which, thanks to its ability to produce acetate and aromatic lactic acids, reduces the phenomena of excessive intestinal permeability while polarizing the immune response towards an arrangement mainly governed by Treg and Th1, thus disadvantaging the expansion of Th2 clones [3,4]. One factor that is frequently associated with microbiota alterations with a consequent increase in allergic risk is birth by cesarean section [5]. Several hypotheses that ‘link’ birth by cesarean section to intestinal dysbiosis in the first months of life and to subsequent atopic manifestations have found valid foundations in numerous studies and meta-analyses [6,7,8,9,10]. Noteworthy is the observation that, regardless of the mode of delivery at birth, a certain type of dysbiosis with low biodiversity, reduction in the bifidobacterial component, and enrichment in Pseudomonadota is mostly observed in children with an increased risk of allergies [11]. Similarly, it should be noted that in children born by a cesarean section where the dysbiosis either does not manifest itself, is counteracted with breastfeeding, or self-extinguishes, the atopic risk is superimposable to that of children born vaginally [12]. On the one hand, reports like these seem to indicate a causative element in the gut microbiota rather than a correlation with atopic manifestation. On the other hand, numerous observations have shown that in children born vaginally, for whom a lower atopic risk is described, the relative percentage of *Bifidobacterium* and *Bacteroides* of maternal origin is increased in the face of a greater containment of Gram-negative populations [13,14,15,16]. Besides atopy, similar dysbiosis, related to birth by cesarean section, has been described as increasing the risk of childhood obesity as well as respiratory infections and dyspepsia [17,18,19,20,21,22,23]. As described for atopy, the abundant presence of bifidobacteria at the beginning of life is also essential to guarantee an optimal immune response and, therefore, a lower incidence of respiratory infections. The phenomenon is well observed in children repeatedly treated with antibiotics. Antibiotic treatment is a strong inducer of intestinal dysbiosis, resulting in a significant reduction in biodiversity and simultaneous impairment of the share of resident bifidobacteria, in which the immune response to vaccines is reduced in close correlation with the number of antibiotic courses [24]. The phenomenon is also evident in the correlation of the relative percentage of bifidobacteria in the first weeks of life (regardless of whether or not antibiotics have been taken or the mode of delivery at birth) with the humoral and cellular consistency of the immune response: the higher the amount of bifidobacteria, the more complete and more effective the immune response is [25]. The correlation between dysbiosis and reduced immune response is not only clinically but also experimentally evident [26]. Indeed, cells of the immune response resort (also) to the production of acetate by the intestinal microbiota to meet their energy needs during a greater workload, such as during a response to a vaccine or an infection [27]. It is therefore evident that in an infant, the production of acetate, largely due to the presence of bifidobacteria, could be considered as a factor of protection from infection.

When dealing with the issue of neonatal bifidobacteria, the concept of species becomes extremely relevant [28]. In fact, the bifidobacteria capable of correct HMO (human milk oligosaccharide) metabolization would essentially be *B. longum infantis*, *B. bifidum*, and *B. breve* [29]. *B. longum infantis*, less frequent in infants born in economically developed countries, is a potent metabolizer of HMOs [30]. On the one hand, by internalizing these substrates, *B. longum infantis* does not share any products deriving from its degradation with the rest of the bacterial community and is therefore defined as ‘selfish’ [31]. On the other hand, the strains of *B. bifidum* externalize the enzymes responsible for the demolition of HMOs and also involve in these metabolic processes the species *B. breve*, which is incapable of a direct metabolization of HMOs in the absence of *B. bifidum* [32]. Strains of *B. bifidum* are therefore considered the cornerstone of the neonatal microbiota [33]. Among their capabilities and further elements of differentiation from *B. longum infantis* is the degradation of the mucin glycans [33,34]. This feature makes them potentially colonizing, even in the absence of HMO (and therefore in the absence of the mother’s milk) [35,36]. Indeed, *B. bifidum* is significantly reduced in allergic children [37]. Their presence develops a T-regulatory response [38], and their administration in children reduces allergic manifestations and atopic dermatitis [39,40]. One of the most studied strains of *B. bifidum* is PRL2010 [41]. Developed in our country (Italy) as a single-strain probiotic for childhood, we decided to evaluate, in a prospective and controlled way, its ability to reduce the incidence of atopy, respiratory infections, and dyspepsia and improve growth rate in the first year of life, when administered for at least 6 months in babies born by cesarean section. We then attempted to validate its possible causative role by investigating the gut microbiotas of some treated and untreated children, selecting among the enrolled children those exclusively formula fed to avoid human milk bacteria coming from entero-mammary cycling that could affect the results of the gut microbiota analysis [42].

## 2. Materials and Methods

### 2.1. The Study

This is a multicenter, randomized, prospective, and controlled trial conducted between September 2019 and May 2021 in Italy at (i) the Neonatology and Neonatal Intensive Care Unit of Santa Maria Goretti Hospital, Latina; (ii) the Neonatology Unit of G.B. Grassi Hospital, Ostia (Rome); (iii) the Neonatology Unit of S.S. Trinità Hospital, Sora (FR); and (iv) the Neonatology and Neonatal Intensive Care Unit of Santo Spirito Hospital, Pescara. The trial has been registered on www.clinicaltrial.gov on 7 July 2023 (Identifier: NCT05936541) and was approved by the Maternal and Infant Department of Santa Maria Goretti Hospital (Protocol: 04-10/05/2022). This study was performed in accordance with the principles of the Declaration of Helsinki. Written informed consent was obtained from parents before entering this study. All parents were assured that declining to participate in this study or leaving this study at any point would not affect the quality of their treatment and that they would thereafter receive the best care available.

### 2.2. Inclusion and Exclusion Criteria

Inclusion criteria were healthy term infants born by cesarean section who regularly shed meconium within 48 h after birth; infants whose mother did not take probiotics and/or declared any gastrointestinal problems during pregnancy. Exclusion criteria were infant diagnosis of congenital diseases and/or acute illness at enrolment or any condition affecting food intake or metabolism; maternal mental and psychosomatic diseases; maternal administration of antibiotics during pregnancy. During the trial, we excluded those infants who suspended the administration of the probiotic strain for a period exceeding 7 days, those whose supplementation was interrupted before the completion of the agreed 6-month period, or those who were supplemented with a different probiotic.

### 2.3. Randomization and Protocol

Participants were randomly classified into two groups by tossing a coin: group A, the treatment arm, and group B, the control arm. Infants in group A were supplemented daily with the probiotic for the first 6 months of life. Infants in group B did not receive any treatment. Infants in group A were treated with the probiotic for the first time 72 h after delivery. The probiotic was administered once a day, dissolved in 20 mL of fresh water. Infants were evaluated at four time points: 3 days after birth (T0), at 3 months of life (T1), at 6 months of life (T2), and at 12 months of life (T3). At each time point, a questionnaire was given to parents during an outpatient visit. The questionnaire was aimed at collecting data concerning the gestational period, including feeding modality, anthropometric measurement, and clinic history, including the incidence of atopic dermatitis, asthma (phenomena of bronchospasm, wheezing), upper respiratory tract infections (including otitis, pharyngitis, rhinitis, cough), lower respiratory tract infections (bronchiolitis and pneumonia), upper and lower dyspeptic syndrome (including gas colic, reflux, diarrhea, constipation, bloating), and growth pattern. Regarding tolerability, the mean severity score was calculated from a four-point scale on symptoms (0 = none, 1 = mild, 2 = moderate, and 3 = severe). The duration of any event was calculated in days. Product safety was assessed by evaluating general physical health and adverse effect reports. Side effects were defined as any medically untoward event detected during this study, whether (or not) they were caused by using the probiotic. The same questionnaires were administered to both groups. The parents of the children involved in this study were requested to have daily contact with the physicians responsible for the study.

### 2.4. The Tested Product

Infants in group A were supplemented with Bactopral^®^, a single-strain probiotic containing *B. bifidum* PRL2010 (not less than 1 × 10^9^ living cells at the expired date), kindly provided by Pharmextracta S.p.A., Pontenure (Italy) and notified to the Italian Health Authorities in 2018 with document number 113,846. Probiotic administration started 72 h after delivery due to the need to collect stools not affected by treatment.

#### 2.4.1. Gut Microbiota Protocol

In accordance with the provisions of the protocol, having received the written consent of the parents and in total anonymity, 10 formula-fed children were randomly selected per group, and their fecal microbiota was analyzed. In the randomly selected 10 children of group A, the sampling was performed twice: 72 h after birth but before the first administration of the probiotic and at the end of the 6 months of treatment. In the randomly selected 10 children of group B, sampling and relative analysis were performed only once at 6 months after birth. Infant stool samples were collected at T0 and/or at T6 directly from diapers into sterile plastic collection tubes by qualified personnel. Stool samples were immediately frozen at −80 °C and stored until further processing for DNA extraction.

#### 2.4.2. Gut Microbiota Analytical Procedures

DNA was extracted from each sample using the QIAmp DNA stool kit following the manufacturer’s instructions (Qiagen Ltd., Strasse, Germany). Extracted DNA samples were kept at −20 °C until they were used for 16S rRNA and for intergenic ribosomal transcriber space (ITS) analyses. Partial 16S rRNA gene sequences were amplified from extracted DNA using primer pair Probio_Uni (5′-CCTACGGGRSGCAGCAG-3′)/Probio_Rev (5′-ATTACCGCGGCTGCT-3′), which targets the V3 region of the 16S rRNA gene sequence. Illumina adapter overhang nucleotide sequences were then added to the partial 16S rRNA gene-specific amplicons, which in turn were further processed by employing the 16S Metagenomic Sequencing Library Preparation Protocol (Part #15044223 Rev. B; Illumina). Amplifications were carried out using a Veriti^TM^ Thermocycler (Applied Biosystems; Foster City, CA, USA). The integrity of the PCR amplicons was analyzed by electrophoresis on a 2200 TapeStation Instrument (Agilent Technologies; Santa Clara, CA, USA). PCR products obtained following the amplification of a section of the 16S rRNA gene were purified by a magnetic purification step involving Agencourt AMPure XP DNA purification beads (Beckman Coulter Genomics GmbH, Krefeld, Germany) in order to remove primer dimers. The DNA concentration of the amplified sequence library was estimated employing a fluorimetric Qubit quantification system (Life Technologies; Carlsbad, CA, USA). Amplicons were diluted to 4 nM, and 5 µL of each diluted DNA amplicon sample was mixed to prepare the pooled final library. Paired-end sequencing (250 bp × 2) was performed using an Illumina MiSeq sequencer with MiSeq Reagent Kit v3 chemicals-600 cycles (Illumina Inc., San Diego, CA, USA). The FASTQ files were processed using QIIME2 [43] as previously described [44]. Paired-end reads were merged, and quality control implementation allowed the retention of sequences with a length between 140 and 400 bp, mean sequence quality score >25, and with truncation of a sequence at the first base if a low-quality sequence within a rolling 10 bp window was found. Sequences with mismatched forward and/or reverse primers were omitted. 16S rRNA ASVs (amplicon sequence variants) were defined at 100% sequence homology using DADA2 [45], and those with less than 10 sequences were filtered. The biological observation matrix (BIOM) obtained was analyzed by summarize_taxa.py script in order to obtain the relative abundance of each taxonomic group for all samples. All reads were classified to the lowest possible taxonomic rank using QIIME2 and a reference dataset from the SILVA database v. 132 [46]. The microbial richness of the samples (α-diversity) was evaluated through the alpha_rarefaction.py script included in the QIIME2 software suite using default parameters. Partial ITS regions were amplified from extracted DNA by PCR using specifically designed primers [47] and sequenced using the MiSeq (Illumina) platform. Following sequencing, the FASTQ files were processed using a custom script based on the QIIME2 software suite. Paired-end reads were assembled to reconstruct the complete Probio-bif_Uni/Probio-bif_Rev amplicons. Quality control retained sequences with lengths between 100 and 400 bp and mean sequence quality scores of >20, while sequences with homopolymers >7 bp in length and mismatched primers were removed. ITS ASVs were defined at 100% sequence homology using DADA2 [48]. All reads were classified to the lowest possible taxonomic rank using QIIME2 and a reference dataset consisting of an updated version of the bifidobacterial ITS database [47]. The number of reads and relative abundances were determined for each bifidobacterial species in each analyzed sample.

### 2.5. Outcomes

Primary outcomes were (i) to evaluate the safety profile of the probiotic strain, including tolerability and occurrence of specific side effects, and (ii) to evaluate the clinical impact of the probiotic treatment with regard to atopy, respiratory infections, dyspepsia incidence, and growth rate. The secondary outcome was to evaluate the possible evolution of gut microbiota structure as a consequence of the probiotic treatment.

### 2.6. Sample Size Calculation

Based on the primary outcomes established in our protocol, the sample size examined in this study was far greater than the required value for a pilot study such as ours [49].

### 2.7. Statistical Analysis

For the statistical analysis of pathology incidences, we relied on the JMP 17.0.0 program for Mac by SAS Institute Inc. (Cary, NC, USA). For the continuous variables—weight and height—considered in this study, the approximation to the normal distribution of the population was tested with the Anderson–Darling test. As results were asymmetrically distributed, data are expressed as median and interquartile range (IQR), 25th and 75th quartile, and non-parametric tests were used to analyze the differences between patients and controls at T0, T1, T2, and T3. Categorical variables were expressed as frequencies (%), and the differences between the groups were analyzed using the chi-squared (χ^2^) test. A *p* < 0.05 was considered significant. For the statistical analysis of the gut microbiotas, the Wilcoxon test was performed to compare ASV relative abundances across samples. To highlight and characterize the type of microorganisms that populate the gut of the newborns with greater precision, phyla, genera, and bifidobacterial species were studied, and a *p*-value of *p* < 0.05 was considered as statistically significant.

## 3. Results

### 3.1. Trend in Infant Number during This Study and Features of the Enrolled Infants

Following the important screening carried out in the hospitals involved in this study, initial consent to participate in the study was progressively given to 500 parental couples or single parents, representing as many newborns as needed for the study that were candidates to be born by cesarean section. However, in numerous cases, more frequently in the participants belonging to the group that would have been treated with the probiotic, consent was withdrawn shortly before the start of the treatment. The justification most often given was the fear that a commitment to 6 months of daily treatment would perhaps be too demanding. In a smaller number of cases, and for many varied reasons, some babies who were expected to be born by cesarean section were delivered naturally; these infants were then excluded from this study. For these reasons, the number of infants enrolled in this study was 413. The sample comprised 230 males and 183 females. Group A, the treatment arm, was composed of 164 infants, and group B, the control arm, was composed of 249 infants. Due to the progressive exclusion process employed during the trial, 268 infants (49 in the probiotic group and 219 in the control group) completed this study. The progressive reduction trend in the number of infants per group during this study is shown in Figure 1. Reasons for progressive exclusion are reported in Appendix A. Infants’ demographics and clinical features are summarized in Table 1. Gestational age, birth weight, feeding type, and milk formulas were not significantly different in the two groups. The different milk formulas adopted at the enrolment are listed in Appendix A. No difference was found regarding maternal age, weight, parity, and swab positivity for *S. agalactiae* (Appendix A). Children’s features, including birth season and possible pathological events occurring or measured in the first 72 h, were not significantly different between the two groups (Appendix A).

### 3.2. Adverse Events and Probiotic Tolerability

No peculiar and/or severe and/or unexpected pathologies were registered during this study. No child showed intolerance or refused to take the probiotic. Probiotic treatment was well-tolerated, and the recorded adverse events overlapped in the two groups of infants both for type (constipation, diarrhea, flatulence, bloating, regurgitation), incidence (about 10% of each group), severity (mild and transient), and duration (3–5 days each) (Appendix A).

### 3.3. Atopic Dermatitis Incidence in the Two Study Groups

Evidence of atopic dermatitis was evaluated clinically. As shown in Table 2, probiotic supplementation significantly reduced the incidence of atopic dermatitis both at T1 (*p* < 0.0001) with 12 infants in group A (7.32%) versus 80 infants in group B (32.13%), at T2 (*p* = 0.0005), with 8 infants in group A (5.84%) versus 44 infants in group B (18.97%), and at T3 (*p* = 0.0130) with 0 infants in group A (0%) versus 25 infants in group B (11.42%). As family history of atopy or other allergic diseases was not considered in the exclusion criteria, we have then analyzed the possible impact on the results of mother atopy. As demonstrated in Appendix A, where the number of mothers with atopy and the number of newborns born by mothers with atopy are shown, both including and excluding mothers with atopy, the statistical analysis of the results did not change. To face atopy, family pediatricians evaluated possible changes in mothers’ diets in the case of breastfed infants and the replacement to a hypoallergenic formula enriched in proteins hydrolyzed into amino acids and peptides or constituted by soy milk in the case of breastfed infants. When these approaches did not produce any significant effect, as well as using emollient creams to keep the skin hydrated and bathing to alleviate itching, family pediatricians prescribed the topical use of corticosteroids. The total prescription rate was significantly different in the two groups (Appendix A), with 6 infants in group A and 35 in group B at T1 (*p* < 0.01), 4 infants in group A and 21 in group B at T2, and 18 infants in group B at T3.

### 3.4. Upper and Lower Respiratory Tract Infection Incidence in the Two Study Groups

Concerning upper respiratory tract infections, probiotic supplementation significantly reduced the incidence both at T1 (*p* < 0.0001), with 22 infants in group A (13.41%) versus 78 infants in group B (31.33%), and at T2 (*p* = 0.0027), with 26 infants in group A (18.98%) versus 78 infants in group B (31.33%) and at T3 (*p* < 0.0001), with 2 infants in group A (4.08%) versus 70 infants in group B 70 (31.96%). Probiotic supplementation also significantly reduced the incidence of lower respiratory infections at T1 (*p* = 0.0112), with 2 infants in group A (1.22%) versus 16 infants in group B (6.43%) and at T2 (*p* = 0.0182) with 1 infant in group A (0.74%) versus 13 infants in group B (5.58%). No difference was observed between the two groups at T3 (Table 2). The different and significant incidence observed in the two groups in terms of respiratory tract infections was also observed in terms of the number of antibiotic prescriptions, with a significantly higher rate of antibiotic prescriptions observed in the control group. Anyway, although in many cases, parents preferred to have access to the pediatric emergency room, only in a limited number of situations did they have to resort to hospitalization. In these cases, however, the duration of hospitalization was never more than 5 days (Appendix A).

### 3.5. Dyspeptic Syndromes Incidence in the Two Study Groups

Only clinical evaluations were conducted for dyspeptic syndrome. Probiotic supplementation was shown to significantly reduce the symptoms of dyspepsia at T1 (*p* < 0.0001) with 59 infants in group A (35.98%) versus 142 infants in group B 142 (57.03%) and at T2 (*p* < 0.0001) with 11 infants in group A (8.03%) versus 74 infants in group B (31.76%). No significant difference was observed between the two groups at T3 (Table 2). If conservative measures like the reduction in the volume of feedings and/or the adoption of correct burping and positioning did not address the problem, family pediatricians adopted different approaches to deal with dyspeptic syndrome in newborns, essentially favoring changes in the maternal diet in the case of breastfed newborns or changes in the replacement of milk formula for non-breastfed children. Regarding maternal diet, in some cases, the change concerned the adoption of a cow-derived food-free diet. Regarding formulas, changes were mainly made in favor of hypoallergenic and protein-hydrolyzed milk, soy milk, and lactose-free milk, with proteins coming from soybeans and with vegetable oils added to provide fatty acids. Only in very few cases, the pediatrician opted for drug therapy with proton pump inhibitors or with histamine-2-receptor antagonists. As reported in Appendix A, the differences in the approaches or in the therapies adopted essentially reflect the different incidence of symptoms and, within certain limits, retain their statistical significance.

### 3.6. Growth Rates in the Two Groups

Analysis of anthropometric data of the enrolled infants during this study showed that after 3 months of treatment, the infants of the probiotic group reported a median weight that was significantly greater (*p* < 0.0001) than reported for the controls (Table 3).

### 3.7. Impact of Probiotic Treatment in the Gut Microbiota

According to the protocol, 10 infants of group A, randomly selected among those whose mothers did not intend to breastfeed and whose stools were sampled 72 h after birth and after 6 months of treatment with the probiotic, were subjected to gut microbiota analysis. As shown in Figure 2, in Group A, α-biodiversity (calculated as the number of ASVs) significantly (*p* < 0.01) increased from 25.56 ± 9.61 (median value: 23) to 50.78 ± 8.57 (median value: 50). At the *phylum* level, Bacteroidota (former Bacteroidetes) significantly increased from about 6% to about 34% (*p* < 0.01), Actinomycetota (former Actinobacteria) significantly increased from about 1% to 7% (*p* < 0.01) and Pseudomonadota significantly decreased from about 53% to about 19% (*p* < 0.01). Regarding group B, we had the chance to analyze 10 infants at 6 months of life. As shown in Figure 3, a significant difference (*p* < 0.05) concerned the *phylum* Pseudomonadota (decreased from about 35% in controls to about 19% in the probiotic group). A tendency, not significant, was also observed in the *phyla* Actinomycetota and Bacteroidota, both of which were more expressed in the group treated with the probiotic versus controls. α-biodiversity at 6 months was 44.20 ± 17.48 in group B (median value: 46.5), which is lower, albeit not significantly, in comparison to the value observed at 6 months in the probiotic group. Regarding bacterial genera (Table 4) in group A, *Bifidobacterium* significantly increased from 0.61 ± 1.75 (median value: 0) observed 72 h after birth to 6.71 ± 8.21 (median value: 2.27) after 6 months of treatment with the probiotic (*p* < 0.01). In group A, the genus *Bacteroides* significantly increased from 5.76 ± 16.42 (median value: 0.19) as observed 72 h after birth to 22.60 ± 26.85 (median value: 7.76) after 6 months of treatment with the probiotic (*p* < 0.01). In group B, at 6 months, the values of *Bifidobacterium* and *Bacteroides* were 2.34 ± 7.43 and 12.57 ± 23.95 (*p* < 0.05), respectively. Regarding Pseudomonadota, the global reduction observed was significantly concerned with the genera *Enterobacter*, *Klebsiella*, and *Citrobacter*, which individually reduced from enrolment to T2 (6 months) by about 90%. In contrast, from enrolment to T2, *Escherichia*/*Shigella* significantly increased (*p* < 0.01) from 4.68 ± 11.56 (median value: 0.02) to 13.82 ± 11.36 (median value: 13.06). In group B, *Enterobacter*, *Klebsiella*, and *Citrobacter* were significantly higher (*p* < 0.05) than in group A at T2. Regarding the genomic analyses of the bifidobacterial species made using ITS, the values of which must be considered by setting the relative value observed for the genus *Bifidobacterium* equal to 100, *B. bifidum* significantly increased (*p* < 0.01) in group A from 0.45 ± 1.06 (median value: 0) to 7.94 ± 13.30 (median value: 2.62) and *B. breve* significantly increased (*p* < 0.05) from 12.44 ± 32.99 (median value: 2.51) to 28.53 ± 40.17 (median value: 7.53). At T2 (6 months) in group B (untreated infants), *B. bifidum* was significantly (*p* < 0.05) expressed at lower values (3.51 ± 4.48; median value: 1.43) while *B. breve*, lower as well, was not significantly different from what was observed at 6 months in the treated infants.

## 4. Discussion

This study demonstrated that 6 months of daily administration of a single-strain probiotic containing *B. bifidum* significantly reduces the incidence of atopic dermatitis, upper and lower respiratory tract infections, and dyspeptic syndrome in infants born by cesarean delivery. Even if a phenomenon of direct causality cannot be certified in any way with the available data, the strong correlation between the use of the probiotic being studied and a lower incidence of disorders is still evident.

Our findings would seem to corroborate the link between dysbiosis and the risk of atopic dermatitis and respiratory infections, demonstrating that supplementing infants with peculiar bifidobacterial species could reduce the incidence of their manifestation at least during the first year of life [50,51]. Atopic dermatitis is considered the first step of the so-called ‘atopic march’ which, due to the concomitant sensitization to foods and aeroallergens in early childhood, predispose children to asthma and allergic rhinitis in late childhood and adulthood [52,53].

*Bifidobacterium* strains are described to influence immune function through toll-like receptors (TLRs), identified as critical for reducing the risk of immunologically mediated diseases, such as allergic diseases [54,55]. The PandA study [56] highlighted how the prevention of atopic eczema by perinatal administration of probiotics occurred through TLRs and, according to Mansfield’s meta-analysis, *Bifidobacterium* strains are the most effective in protecting infants against eczema [57]. It is, therefore, possible to hypothesize that the use of probiotic supplements can modify the composition of the intestinal flora of children, subsequently modulating the reactivity of the immune system and possibly playing an important role in the prevention of atopic dermatitis. We also observed a lower incidence of respiratory infection in the group who took the probiotic than those in the control group. This is in line with recent findings that demonstrate that the percentage amount of *Bifidobacterium* spp. in children born via vaginal delivery is inversely correlated with the number of respiratory tract infections [14,58].

To exclude possible biases that could affect the results, we assessed the possible impact of several confounding parameters that may have affected the significance of our results. Yet, as shown in Table 1 and Appendix A, we did not detect significantly different parameters in the two groups of children. At the time of writing the “Discussion” paragraph, we retrospectively evaluated other parameters not reported in Appendix A, such as the Apgar score at birth, the number of brothers and sisters of the children enrolled, the possible exposure to cigarette smoking, the socio-economic and the cultural level and of the parents, the residence (countryside or city) of the families participating in this study and the presence of pets or proximity to farm animals. Anyway, no parameter considered was found to be significantly different in the two groups. Finally, due to the COVID-19 pandemic, no children had the opportunity to attend the usual community places such as nurseries, as much of this study took place during the mandatory lockdown period. In our opinion, therefore, the parameter that most likely determined a different number of diagnoses of atopy and respiratory infections, at least considering those parameters that we can indeed control, was the treatment with the probiotic and its impact on microbiota structure.

Regarding the impact of the microbiota structure on the host’s wellness, some recent findings suggest that MHC (major histocompatibility complex) could promote a benign host–microbiota symbiosis, also reducing the Th2-polarized immune response and infections [59,60,61,62,63]. As we did not evaluate the MHC, we cannot establish at all if this parameter affected our results. In any case, the randomization adopted and the overlap of the characteristics describing the two groups under study allowed us to hypothesize that the relevance of the MHC may have been spread across both groups, thus not constituting an important bias.

Concerning the positive results recorded for dyspeptic syndrome, our assumption is that the probiotic supplementation could have stabilized the gut microbiota, reducing the variability associated with the commencement of weaning, with a consequent reduction in dyspeptic phenomena. In fact, in the first months of life, the gut microbiota is less stable than in late childhood, and changes in phylogenetic diversity may occur [64,65]. Weaning causes progressive changes in the composition of the microbiota until the expression of genes that contribute to anti-microbial immunity, including inflammatory molecules such as tumor necrosis factor alpha and interferon-γ [66]. Unfortunately, due to the lockdown for COVID-19, we evaluated the incidence of dyspeptic syndrome only by the clinical evidence of symptoms and/or by questionnaire. Not having the possibility, for instance, to evaluate the gastric pH in case of gastric reflux, it could be that the importance of our observation is reduced.

The lower weight in the first trimester observed in infants who were not given the probiotic could be related to the higher incidence of respiratory infections and dyspeptic syndrome that can contribute to a reduction in static-weight growth. Indeed, a transient lack of appetite could be a direct consequence of the disease. Furthermore, a recent report has demonstrated that gut microbiota influences infant growth, especially in the first weeks of life [67]. Finally, a well-balanced intestinal microbiota can better protect the gut and help it perform its functions optimally, including the absorption of all the nutrients necessary for the growth of the infant.

Most of the reflections set out above find support in the microbiota analyses we performed on 20 formula-fed newborns (infants of both groups were fed with the same types of milk formula, which included the same dosage of 2-fucosyl-lactose), none of which were ever breastfed and 10 of which were not treated with any probiotics. As the results demonstrate, treatment with the *B. bifidum* strain causes a shift in the microbiota of the treated infants towards greater eubiosis determined by an increase in the value of the genus *Bifidobacterium* and of the *B. bifidum* species. The colonization of the latter would also seem to confer a proliferative advantage, through a well-demonstrated cross-feeding process, to the species *B. breve* [68]. Beyond the bifidobacterial component, the microbiota of the treated infants improves in overall terms, demonstrating higher biodiversity, the increase in *phylum* Bacteroidota and the genus *Bacteroides*, and the decrease in Pseudomonadota, with reference to genera considered putative pathogens such as *Enterobacter*, *Klebsiella*, and *Citrobacter* [69].

Our results have been obtained by administering a well-investigated strain of *B. bifidum*, which had already demonstrated gut colonizing and pro-eubiotic capability in experimental models [70]. This strain, indeed, an intriguing example of a microbial inhabitant of gut microbiota, is currently considered to have a clear ecological role within the infant gut. This is mainly due to its genetic adaptation to the human gut microbiota, its interactive role with the host’s immune system, and its activity in preserving mucosal integrity [71,72,73,74,75].

The strain PRL 2010 has also shown its safety profile. For the entire duration of this study, and certainly during the 6 months in which it was administered daily, it demonstrated a notable gastrointestinal safety profile (Appendix A). In fact, in the treated group, no peculiar adverse events were highlighted that were different from those observed in the control group. Similarly, however, the probiotic strain does not seem to have reduced the (few) gastrointestinal manifestations that we observed in the study subjects. However, we observed that some disorders, such as bloating and flatulence for example, were reduced in those subjects in whom, due to dyspepsia, the family pediatrician had proposed changes in the maternal diet (in the case of breastfed children) or the use of hypoallergenic milk (in the case of formula-fed children). It is, therefore, possible that some of these children also had problems such as lactose intolerance.

Due to the many limitations of this study, our findings cannot be considered conclusive. They have been deduced from a study that was not performed in double-blind conditions and not controlled with a placebo. Moreover, the two groups were not perfectly balanced, with fewer infants in the treated group, and the clinical signs of pathology have been reported without analyzing specific parameters like, for instance, blood tests. Furthermore, the analyses of the microbiota were performed on a very small number of infants and only in two moments during this study (T = 72 h and T = 6 months), and these two aspects may have failed to allow the observation of all the consequences that supplementation with the probiotic strain can produce. However, the greatest limitation attributable to the analytical part was the unavailability of the fecal samplings of the control group performed at T0. In the initial design of this study, which envisaged only the evaluation of the incidence of atopy, respiratory infections, and dyspepsia, as well as monitoring of the growth curves of the enrolled newborns, the intention was to verify the microbiota of only some of the treated subjects. The objective of this analysis, perhaps modest, was the verification of the effective gut colonization of the bifidobacterial strain used. The analyses of the 16S rRNA and ITS genes do not allow the identification at the strain level but only at the genus and species levels. We, therefore, decided to partially modify the study drawing and to include sampling the stool from the controls. This sampling, however, was only possible at 6 months. That said, it appears quite evident from the comparison at 6 months between the two groups that the group treated with the probiotic demonstrated a better eubiosis, with significant differences in bifidobacteria (mostly present in the treated group) and Enterobacteriaceae (mostly present in the control group) and with a better profile regarding richness and Bacteroidota.

## 5. Conclusions

Despite some important limitations, our randomized, prospective, and controlled study has demonstrated a positive clinical role (better growth rate in the first trimester, lower incidence of atopy, respiratory infection, and dyspepsia) exerted by supplementation with *B. bifidum* PRL2010 in infants delivered by cesarean section. It is worth noting that some of these effects have also been observed during the 6 months of follow-up. This study has also highlighted the good safety profile of the nutritional supplement used. Notably, to our knowledge, this is the first prospective study analyzing the role played by a probiotic product containing only one specific strain of the species *B. bifidum* in newborns born by cesarean section in which the aim was to detect, among others, a possible anti-atopy effect. Other studies attempted to observe similar effects, but either the authors tested a mixture of probiotics (including or not a strain of *B. bifidum*), or they did not enroll children born at term by cesarean section [39,76,77,78,79,80,81,82,83,84,85,86,87,88,89,90,91,92,93,94,95,96,97]. Despite the differences in the type of enrolled children and/or in the probiotics tested, our results, to some extent, confirm what has been described earlier. That is, (1) *Bifidobacterium* is an inhabiting and beneficial genus for newborns; (2) it likely promotes an anti-atopic and Th1-polarized immune response, especially in cesarean-born children; (3) it is capable of counteracting Enterobacteriaceae massive relative presence, likely reducing the risk of LPS-driven inflammatory responses; (4) generally, it promotes infant well-being; and (5) the species *B. bifidum* is likely a possible key player of the infant gut microbiota. Finally, to our knowledge, this is the first time this strain has been clinically verified for safety and efficacy, independent of the outcomes established. According to our results, it may be advisable to recommend the administration of this probiotic strain in newborns delivered by cesarean section and at risk of dysbiosis. Further clinical trials (double-blind and placebo-controlled) are about to be designed to confirm the validity and reproducibility of our results.

## Figures and Tables

**Figure 1 microorganisms-12-01093-f001:** Infants enrolled and lost during follow-up. All patients who did not comply with the protocol were progressively excluded from this study. The reasons for the exclusions are reported in Appendix A.

**Figure 2 microorganisms-12-01093-f002:** Graphic representations at phylum level of gut microbiota of infants of the probiotic-treated group (Group A) at enrolment (**left**) and after 6 months of treatment (**right**). Significant differences concern the phyla Actinomycetota (former Actinobacteria) and Bacteroidota (former Bacteroidetes), which both increased after 6 months of treatment versus enrolment, and Pseudomonadota (former Proteobacteria), which reduced at 6 months versus enrolment. Verrucomicrobiota, Mycoplasmatota, Fusobacteriota, and Bacillota are the new phyla names, respectively, for Verrucomicrobia, Tenericutes, Fusobacteria, and Firmicutes. ASV: amplicon sequence variant.

**Figure 3 microorganisms-12-01093-f003:** Graphic representations at phylum level of gut microbiota of infants of the control group (Group B) at 6 months of life (**left**) and of the probiotic group (Group A) after 6 months of treatment (**right**). Significant differences concern the phylum Pseudomonadota (former Proteobacteria), which reduced in the probiotic group versus control. A tendential but not significant difference is also observed in the phyla Actinomycetota (former Actinobacteria) and Bacteroidota (former Bacteroidetes), and in biodiversity, both are more expressed in the group treated with the probiotic versus controls. Verrucomicrobiota, Mycoplasmatota, Fusobacteriota, and Bacillota are the new phyla names, respectively, for Verrucomicrobia, Tenericutes, Fusobacteria, and Firmicutes. ASV: amplicon sequence variant.

**Table 1 microorganisms-12-01093-t001:** Infants’ features at the enrolment.

	Group AProbiotic-Treated	Group BControl Group
Enrolled subjects: N	164	249
Gender: M/F	97/67	133/116
Gestational Age	38.4 ± 3.1	38.4 ± 2.7
Birth weight[IQR]	3.310 ± 0.573[0.557]	3.280 ± 0.481[0.640]
Feeding: formula/breast	72/92	105/144

N: number; M: male; F: female. For gestational age and birth weight, values are expressed as median ± standard deviation. IQR: interquartile range.

**Table 2 microorganisms-12-01093-t002:** Incidence of atopic dermatitis, upper and lower respiratory tract infection, and dyspeptic syndrome at 3, 6, and 12 months.

Clinical Characteristic	Group A: n (%)	Group B: n (%)	*p*-Value
**Atopic dermatitis**			
3 months (T1)	12 (7.32)	80 (32.13)	<0.0001
6 months (T2)	8 (5.84)	44 (18.97)	0.0005
12 months (T3)	0 (0)	25 (11.42)	0.0130
**Upper respiratory tract infections**			
3 months (T1)	22 (13.41)	78 (31.33)	<0.0001
6 months (T2)	26 (18.98)	78 (33.48)	0.0027
12 months (T3)	2 (4.08)	70 (31.96)	<0.0001
**Lower respiratory tract infections**			
3 months (T1)	2 (1.22)	16 (6.43)	0.0112
6 months (T2)	1 (0.73)	13 (5.58)	0.0182
12 months (T3)	0 (0)	0 (0)	N. S.
**Dyspeptic syndrome**			
3 months (T1)	59 (35.98)	142 (57.03)	<0.0001
6 months (T2)	11 (8.03)	74 (31.76)	<0.0001
12 months (T3)	0 (0)	0 (0)	N. S.

Group A: probiotic treated. Group B: control group. n: number of subjects. N. S.: not significant.

**Table 3 microorganisms-12-01093-t003:** Weight gain (in grams) at 3, 6, and 12 months.

Weight	Group A (Probiotic-Treated)	Group B (Control Group)	*p*-Value
3 months (T1)	6037.1 ± 797.8 [6000; 5500–6500]	5548.9 ± 831.0 [5500; 5000–6015]	<0.0001
6 months (T2)	7823.6 ± 940.3 [7810; 7220–8350]	7819.5 ± 1042.5 [7800; 7000–8300]	N. S.
12 months (T3)	10,587.7 ± 1374.5 [10,500; 9800–11,000]	11,062.7 ± 1499.7 [10,800; 10,000–12,000]	N. S.

Median and range values are expressed in square brackets. N. S.: not significant.

**Table 4 microorganisms-12-01093-t004:** Bacterial genera and species in Group A (at T0 and T2) and in Group B (at T2).

	Group A at T0	Group A at T2	Group B at T2
*Bacteroides*	5.76 ± 16.42	22.60 ± 26.85 *	12.57 ± 23.95 °
*Bifidobacterium*	0.61 ± 1.75	6.71 ± 8.21 *	2.34 ± 7.43 °
*B. bifidum*	0.45 ± 1.06	7.94 ± 13.30 *	3.51 ± 4.48 °
*B. breve*	12.44 ± 32.99	28.53 ± 40.17 *	17.93 ± 38.33
*Escherichia/Shigella*	4.68 ± 11.56	13.82 ± 11.36 *	19.06 ± 30.51
*Enterobacter*	13.51 ± 21.70	1.34 ± 1.55 *	3.08 ± 6.11 °
*Klebsiella*	20.20 ± 40.27	0.92 ± 1.64 *	7.33 ± 12.44 °
*Citrobacter*	5.44 ± 10.08	0.07 ± 0.10 *	2.09 ± 0.21 °

Group A: probiotic-treated infants. Group B: control group. T0: 72 h after birth. T2: 6 months. Values are expressed as mean ± standard deviation. * *p* < 0.01 versus Group A at T0. ° *p* < 0.05 versus Group A at T2.

## Data Availability

The raw data supporting the conclusions of this article will be made available by the authors on request.

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
