# Peer review of "Effect of Bifidobacterium bifidum Supplementation in Newborns Born from Cesarean Section on Atopy, Respiratory Tract Infections, and Dyspeptic Syndromes: A Multicenter, Randomized, and Controlled Clinical Trial"

_microorganisms, 2024, doi:10.3390/microorganisms12061093_

Round 1

Reviewer 1 Report (Previous Reviewer 2)

Comments and Suggestions for Authors

The manuscript by Anna Rita Bellomo and colleagues entitled "Effect of Bifidobacterium bifidum Supplementation in Newborns Born from Caesarean Section on Atopy, Respiratory Tract Infections and Dyspeptic Syndromes: A Multicentre, Randomized and Controlled Clinical Trial" is a resubmission to Microorganism.  Regarding the present manuscript, I would like to make a few comments.

  • Please select American or British English, not both. In the titles, they use multicentre, British, and randomized, American, so please review the entire document, especially the word cesarean.

  • In the introduction, the authors use italic phylums and do not use italics for the other phylas. It may be possible for the authors to use the new names for phyla, for example Bacillota for Firmicutes.

  • Is there a missing value in the sample size, how large is the standard deviation, what is the mean of effect?

  • Please modify Figure 1 as it is in a very low resolution. Please modify all the tables as they are out of style

  • To avoid 0.00 in figures 2 and 3, the authors could use the variable taxa <1% of relative abundance

  • Thank you for adding the limitation paragraph.

  • It would be useful to highlight the main outcomes found in the present study and compare them with similar studies or meta-analyses.

Comments on the Quality of English Language
  • Please select American or British English, not both. In the titles, they use multicentre, British, and randomized, American, so please review the entire document, especially the word cesarean.

Author Response

Reviewer 2 Report (Previous Reviewer 3)

Comments and Suggestions for Authors

The research objectives are clear and the results are somewhat enlightening.

The author made some modifications and the article was resubmitted to this journal.

Round 2

Reviewer 1 Report (Previous Reviewer 2)

Comments and Suggestions for Authors

Thank you for responding so quickly to my previous comments. I do not need to make any further changes. Now, the manuscript reads smoothly.

This manuscript is a resubmission of an earlier submission. The following is a list of the peer review reports and author responses from that submission.

Round 1

Reviewer 1 Report

Comments and Suggestions for Authors

The manuscript presents important results and explores supplementing infants delivered by cesarean section with Bifidobacterium bifidum PRL2010 to improve their health outcomes. It shows significant reductions in atopic dermatitis, respiratory infections, and dyspeptic symptoms, suggesting probiotic supplementation could mitigate the risks of gut dysbiosis associated with cesarean birth.

Some specific comments:

-          At the start of the abstract there should be one background sentence and what was the problem with previous studies

-          Line 18 – please specify to whom this sentence refers (mother or newborn).

-          Line 38 – please define the term “eubiotic”

-          Line 100 – 101 – “Informed written was obtained from parents before entering in the study.” – please define what was obtained.

-          The materials and method section – especially the subsection “Gut microbiota analysis” should be separated in subsections for better overall visibility

-          Line 145 - Verity Thermocycler (Applied Biosystems) – please specify for every equipment used the type and country of production – revise the whole manuscript

-          Line 318 – recent finding > recent findings

-          Lines 318 – 320 – this sentence should be rephrased.

Comments on the Quality of English Language

The authors should also revise the whole manuscript by a native English speaker, to make it more easily understandable, currently, it is hard to understand some sections.

Author Response

Tx.

Reviewer 2 Report

Comments and Suggestions for Authors

According to the manuscript by Anna Rita Bellomo and colleagues titled "Effect of Bifidobacterium bifidum supplementation in newborns born from caesarean section on atopy, respiratory tract infections and dyspeptic syndromes: a multicentre, randomized and controlled clinical trial". There is some evidence that a Caesarean section may trigger gut dysbiosis. There is evidence that Bifidobacteria, and in particular B bifidum, play a significant role in promoting gut eubiosis in children. As a result, we evaluated the impact of 6 months of supplementation with B bifidum PRL2010 on the incidence of atopy, respiratory tract infections, and dyspeptic syndromes in 164 children born by cesarean (compared with 249 untreated controls) at 1 year of age. In a multicenter, randomized and controlled study, we found that probiotic supplementation significantly reduced the incidence of atopic dermatitis, upper and lower respiratory tract infections, and signs and symptoms of dyspeptic disorders. When compared to control group supplements, B bifidum supplementation significantly increased gut microbiota biodiversity and relative values of the phyla Bacteroidetes and Actinobacteria, the genus Bacteroides, Bifidobacterium, and B bifidum species, but reduced the relative distribution of Escherichia/Shigella and Haemophilus. Supplementation with B bifidum for six months reduces the risk of gut dysbiosis in children born by cesarean section, with the effects visible in the following six months of follow-up as well. Regarding the present manuscript, I would like to make a few remarks.

  • Use the template of the journal and ensure that American English or British English is used, not both.

  • Which protocol or scale was used to determine the safety and adverse events based on the outcomes?

  • Why were so many children excluded from the probiotic arm?

  • It is important to note that the microbial analysis is only a snapshot of the total number of patients that completed the study

  • By providing some correlations between microbe abundance and the study results (safety and adverse events). This provides a better understanding of the findings, in my opinion

Author Response

Tx

Reviewer 3 Report

Comments and Suggestions for Authors

It is of great significance for the author to conduct such a study (Effect of Bifidobacterium bifidum supplementation in newborns born).

Firstly, the study's novelty is limited.  

There are many studies on the effects of probiotics, including bifidobacteria, on the intestinal tract and health. The study's novelty should be enhanced.

Secondly, more comprehensive data analysis should be performed.  

The microbial analysis, for instance, should extend beyond the phylum level to delve into the genus of  microbiome.  Additionally, exploring immunity, metabolism or interactions between the immune system, metabolism, and the microbiome would provide a deeper understanding of the underlying mechanisms.

Finally, the study's focus on nutritional outcomes suggests that a more appropriate venue for publication would be a nutrition or medical journal. These specialized journals would better cater to the study's audience.

In summary, while the study raises intriguing questions about the role of Bifidobacterium bifidum in newborn health, its potential contribution would be amplified by a more in-depth analysis and a targeted publication strategy.

Author Response

Tx.

Round 2

Reviewer 1 Report

Comments and Suggestions for Authors

The authors improved the manuscript considerably. In my opinion, it can be accepted for publication.

Comments on the Quality of English Language

The English is quite good

Reviewer 2 Report

Comments and Suggestions for Authors

I would like to thank the authors for taking the time to respond to my previous comments. There were two important questions raised, the high rate of exclusion on the probiotic arm and the number of patients measured in the microbiota composition. COVID19 is the subject of the first question, while high correlations are the subject of the second question.In my opinion, it is important to evaluate the majority of subjects, rather than just 10 out of more than 200 in the control group and 49 in the probiotic groupThere is also the issue of comparing those numbers, which should be mentioned. As a final note, I would like to point out that there is a template for the preparation of the manuscript, please make use of it.

Reviewer 3 Report

Comments and Suggestions for Authors

It is of great significance for the author to conduct such a study (Effect of Bifidobacterium bifidum supplementation in newborns born).

The author has made some modifications, but the modifications are relatively few regarding the content, and most of the modifications are in the format.

It is recommended that the discussion section be streamlined or highlight key points, such as the functions of bifidobacteria.

It is recommended that the author modify or add documents, such as PMID 34560336 and PMID 35079761.